# Prevalence of Gestational Diabetes in preCOVID-19 and COVID-19 Years and Its Impact on Pregnancy: A 5-Year Retrospective Study

**DOI:** 10.3390/diagnostics12051241

**Published:** 2022-05-16

**Authors:** Sorina Chelu, Elena Bernad, Marius Craina, Radu Neamtu, Adelina Geanina Mocanu, Corina Vernic, Veronica Daniela Chiriac, Larisa Tomescu, Claudia Borza

**Affiliations:** 1Discipline of Pathophysiology, Victor Babes University of Medicine and Pharmacy Timisoara, 300041 Timisoara, Romania; sorina.chelu@yahoo.com (S.C.); borza.claudia@umft.ro (C.B.); 2Discipline of Obstetrics and Gynecology, Victor Babes University of Medicine and Pharmacy Timisoara, 300041 Timisoara, Romania; ebernad@yahoo.com (E.B.); crainamariuslucian@gmail.com (M.C.); radu.neamtu@umft.ro (R.N.); adelinaerimescu@yahoo.com (A.G.M.); tomescu.larisa@umft.ro (L.T.); 3Computer Science and Medical Biostatistics Discipline, Victor Babes University of Medicine and Pharmacy Timisoara, 300041 Timisoara, Romania; cvernic@umft.ro

**Keywords:** gestational diabetes, pregnancy, COVID-19

## Abstract

Gestational diabetes mellitus (GDM) affects a total of 3% to 9% of all pregnancies. It has a high impact on both mother and baby, increases the perinatal risks, and predicts the presence of long-term chronic metabolic complications. The aim of our study is to determine the incidence of GDM in tertiary hospitals in the west part of Romania to lay out the risk factors associated with GDM and to observe the evolution of pregnancy among patients with this pathology by emphasizing the state of birth of the fetus, the birth weight, and the way of birth. We also want to compare the prevalence of GDM in preCOVID-19 (Coronavirus disease) versus COVID-19 years. The study took place between January 2017 and December 2021 at the Municipal Emergency Hospital of Timisoara, Romania. The proportion of births with GDM was significantly increased during the COVID-19 period compared to the preCOVID-19 period (chi2 Fisher exact test, *p* < 0.001). The period 2020–2021 represents a significant risk factor for GDM births (OR = 1.87, with 95% CI = [1.30, 2.67]). COVID years represent a risk period for developing gestational diabetes, which can be explained by reduced physical activity, anxiety, or modified dietary habits, even if the follow-up period was not impacted.

## 1. Introduction

Gestational diabetes mellitus (GDM) is a carbohydrate intolerance resulting in hyperglycemia of variable severity, with onset or first recognition during pregnancy. According to the Romanian Society of Diabetes and Metabolic Diseases, since 2013, GDM has been defined as any type of glucose intolerance with onset or first recognition during pregnancy.

Since 2011, the American Diabetes Association (ADA) has defined GDM as any degree of glucose intolerance with onset after 24–28 weeks of gestation; it is diagnosed by the presence of a single pathological blood glucose value during the oral glucose tolerance test (OGTT) with 75 g of glucose. Due to the obesity epidemic, the number of patients with type 2 diabetes has increased among young women of childbearing age, and the number of pregnant women with previously undiagnosed type 2 diabetes has increased. Any definition must take the perinatal morbidity and mortality into consideration, the mother’s risk of developing further diabetes, and the fetus’ risk of having a metabolic pathology later in adulthood. Pregnancies affected by GDM impose a risk of cesarean and operative vaginal delivery due to macrosomia and shoulder dystocia [1,2].

Risk factors for GDM include the following: overweight and obesity, previous GDM or prediabetes, diabetes in an immediate family member, previously delivering a baby weighing more than four kilograms, and the advanced age of pregnant women. The correlation between diabetes and the age of a pregnant woman is linear, the prevalence of diabetes increasing with age [3,4].

The prevalence of GDM varies considerably due to the use of the data source of the evaluated population and the diagnostic criteria [5,6]. The incidence is between 0.15% and 15% of pregnancies [5,6]. In low- and middle-income countries, the incidence is estimated at around 15% of pregnant women [6,7,8,9].

## 2. Materials and Methods

### 2.1. Objects and Collection

The study was conducted retrospectively in a single-center study, a tertiary referral center, Municipal Emergency Hospital of Timisoara (SCMUT), between January 2017 and December 2021. The information was obtained by analyzing the patient’s observation sheets, birth records, and birth and operation protocols.

The study screened 2 groups of patients: the 1 group consisted of all pregnant women that were diagnosed with gestational diabetes and gave birth during the period 2017–2021 in the Municipal Emergency Clinical Hospital Timisoara (SCMUT); the second group consisted of the total number of pregnant women who gave birth during the period 2017–2021 in SCMUT and did not have GDM.

The criteria for inclusion in the first group were: pregnant women who meet the gestational diabetes mellitus diagnostic criteria according to ADA (American Diabetes Association) recommendations by performing a 75 g TTGO with plasma glucose measurement when the patient is fasting and after 1 and 2 h. The diagnostic of gestational diabetes mellitus is given when any of the following is met or exceeded: fasting—92 mg/dL (5.1 mmol/L);1 h—180 mg/dL (10.0 mmol/L); 2 h—153 mg/dL (8.5 mmol/L). TTGO was performed in cases where the anamnesis showed the presence of clinical risk factors or elevated blood glucose levels. The gestational age at which TTGO was applied ranged from 24 to 32 weeks of gestation. An early screening was applied to patients in which risk factors, such as obesity, impaired carbohydrate metabolism, or even a history of GDM, were identified. The exclusion criteria were pregnant women who did not have TTGO or had normal TTGO results and patients with Type 1 diabetes and Type 2 diabetes.

The criteria for inclusion in the second group were: all the pregnant women who were hospitalized for childbirth assistance between 2017 and 2021 in our Hospital (SCMUT) and had a normal TTGO result. The exclusion criterion in the second group was pregnancy associated with GDM.

The study followed the prevalence of pregnancies with gestational diabetes mellitus between 2017 and 2021 in our hospital compared to the total number of pregnancies screened for GDM in that period, the prevalence of pregnancies with gestational diabetes mellitus in preCOVID years (2017, 2018, 2019) and COVID years (2020, 2021), the prevalence of risk factors for gestational diabetes: mother’s age, obesity, in the group of pregnancies with gestational diabetes mellitus compared with the group of pregnancies without GDM.

In the group of pregnancies with GDM, we analyzed the way of birth (natural delivery versus cesarean delivery), the fetus status at birth, as reflected by the Apgar Score, and the incidence of fetal malformations in this group.

The selection of patients for the study is described in the flowchart in Figure 1.

### 2.2. Procedures

Gestational diabetes mellitus diagnostic criteria, according to ADA (American Diabetes Association) recommendations by performing TTGO, consist of oral administration of 75 g glucose with plasma glucose measurement when the patient is fasting and after 1 and 2 h. The diagnostic of GDM is made when any of the following is met or exceeded: fasting—92 mg/dL (5.1 mmol/L); 1 h—180 mg/dL (10.0 mmol/L); 2 h—153 mg/dL (8.5 mmol/L) [10].

### 2.3. Statistical Analysis

Statistical data was processed using SPSSv17. The numerical variables with normal distribution are represented by mean ± standard deviation, and for comparison between two sets of these kinds of values, the unpaired *t*-test was used; the numerical variables without normal distribution are represented by the median (interquartile range), and, to make a comparison between them, the Mann–Whitney U-test was applied. We used the Shapiro–Wilk test for normality testing. The chi2 test was used for the comparison between nominal variables, and the risk analysis was utilized in some cases. The results were considered significant at a value of *p* < 0.05.

## 3. Results

Between January 2017 and December 2021, 17,230 births took place at our hospital (SCMUT). Of these, 2110 mothers performed TTGO, and 122 were births from mothers with gestational diabetes mellitus, which represents 5.78%.

The distribution per year of the prevalence of GDM births is presented in Table 1, where we can see that the highest incidence of births with GDM was in 2021 (8.48%) and the lowest incidence was in the year 2017 (2.77%).

The study also analyzed the prevalence of GDM births in preCOVID-19 years (2017, 2018, 2019) and COVID-19 years (2020, 2021) in order to see if it was any difference in the prevalence of GDM between these two periods

In the preCOVID years (2017, 2018, 2019), the prevalence of GDM was 4.06% out of a total of 1232 births, and in the COVID years (2020, 2021), the incidence was higher (8.20%) out of a total of 878 births.

The proportion of births with GDM was significantly increased during the COVID-19 period compared to the preCOVID-19 period (chi2 Fisher exact test, *p* < 0.001).

The COVID-19 period represents a significant risk factor for GDM (OR = 1.87, with 95% CI = [1.30, 2.67]).

The prevalence of GDM is higher in COVID-19 years (2020–2021). If we compare the incidence in 2018 with the incidence in 2021, we can observe that the incidence of GDM was 2.91% in 2018 and 8.48% in 2021 (a COVID year).

We analyzed two important risk factors for gestational diabetes mellitus: the groups’ age and obesity, and we compared the incidence in the group of mothers with GDM and the group of mothers without GDM. We also analyzed the effect of GDM on the fetus by assessing the incidence of fetal malformation, the weight of the fetus at birth (noting the incidence of macrosomal fetuses), the way of birth, and the Apgar Score (which reflects the condition of fetuses at birth).

The characteristics studied in the GDM group are presented in Table 2 and Table 3.

We observed that the highest incidence of diabetes out of a total of 122 births with GDM was in the age group of 31–40 years (62.30%), which is in agreement with the data in the literature [10,11], and the lowest incidence was in the group >40 years (1.64%).

The proportion of mothers with GDM is significantly increased among mothers over the age of 30 (1.04%) compared to those who are under 30 years (0.45)—chi2 Fisher exact Test, *p* < 0.001.

Mothers over the age of 30 represent a significant risk factor for the occurrence of GDM (OR = 2.32, with 95% CI = [1.61, 3.37]).

We tried to see if there was any correlation between maternal obesity and the incidence of diabetes. The proportion of mothers with GDM is significantly increased among those with obesity (54.62%) compared with those without obesity (45.38%) (chi2 Fisher exact test, *p* < 0.001. Maternal obesity represents a significant risk for GDM (OR = 402.29, with 95% CI = [258.74, 625.46]).

Another important aspect we studied was fetus birth weight in order to see the incidence of fetal macrosomia. The incidence of fetal macrosomia in the group of 122 pregnancies with GDM was increased. The increased risk of macrosomia is mainly due to the increased insulin resistance of the mother.

The proportion of macrocosmic newborns is significantly increased among mothers with GDM (25.41%) compared to the proportion of mothers without GDM (3.85%) (chi^2^ Fisher exact test, *p* < 0.001).

GDM is a significant risk factor for macrocosmic fetus (OR = 8.52, with 95% CI = [5.62, 12.90]).

We evaluated in our study the way of birth in the group of gestational diabetes mellitus pregnancies, and we observed that most pregnancies were delivered by cesarean section (86.89%), with 13.11% delivered naturally. The main indications for C-section were fetal macrosomia, protracted second stage of labor, fetal distress reflected in the Apgar Score, and also scarred uterus after previous cesarean sections. We did not have operative deliveries (vacuum or forceps).

We evaluated the condition of the fetus at birth by Apgar Score, and we found the following results: in 11 cases, the Apgar Score was 6 (9.016%); in 20 cases, the Apgar Score was 7 (21.311%); and in 85 (69.872%) cases, the Apgar Score was 9.

The incidence of fetal malformations among mothers with gestational diabetes happened to be somewhere around 6% [12,13,14,15,16]. In our retrospective study, we found 2 cases of fetal malformations out of a total of 122 births of mothers with gestational diabetes mellitus, which is a percentage of 1.639%.

We also compared all the used statistical tests and obtained these results: the value of hemoglobins in preCOVID patients was 11.18 ± 1.48 and in pregnant women during COVID 10.92 ± 1.38; the preCOVID hematocrit value was 34.1 (7.44), and in the COVID period, (6.7); the level of blood sugar (mg/dL) preCOVID was 103 (35.5), and in the period of COVID, 101.5 (32.75). Additionally, the weight of the fetus in the preCOVID period, on average, was 3400, and in the period of COVID, 3575; the Apgar index in both periods was 9; in the preCOVID period, 17 (34%) patients had HTAIS, with 19 (26.4%) patients in the COVID period; in the preCOVID period, there were 41 (82%) newborns with pathology, and in the COVID period, 50 (69.4%).

Hemoglobin, hematocrit, leukocyte, and blood sugar levels were insignificantly low in the COVID versus preCOVID period; birth weight was insignificantly increased in the COVID period, and the proportions of HTAIS and the pathology associated with the newborns were insignificantly low in the COVID period (*p*-value meanings and the used tests are listed in Table 4).

## 4. Discussion

Gestational diabetes mellitus is one of the leading causes of mortality and morbidity for both mother and child [17,18,19,20]. Newborns from mothers with GDM are at risk of being macrosomic, may suffer from more congenital abnormalities, and have a greater risk of developing neonatal hypoglycemia and Type 2 diabetes later in life. Maternal gestational diabetes exposes the fetus to higher concentrations of glucose than normal, which forces the fetus to increase its own insulin production. This situation can cause the fetus to grow excessively, a condition known as large for gestational age. A fetus with a birthweight exceeding 4000–4500 g is referred to as a macrocosmic fetus. Mothers with GDM are at risk of developing gestational hypertension or preeclampsia and have an increased risk of cesarean delivery. COVID-19 during pregnancy is strongly associated with preeclampsia, especially among nulliparous women [21].

According to the consensus and expert opinion in the ACOG guideline on GDM, screening for GDM usually occurs at 24–28 weeks of gestation, but early screening is recommended. In our study, we screened for GDM only in selected cases.

Gestational diabetes mellitus is frequently described as the most common metabolic disorder of pregnancy, with its prevalence increasing at epidemic proportions. Other studies have described that the incidence of GDM varies considerably due to the use of different data sources and is between 0.15% and 15% of pregnancies [22,23,24]. However, an accurate estimation of the global incidence of gestational diabetes does not exist because of the lack of uniform standards in glucose tolerance testing around the world.

The incidence of pregnancies with GDM in our study group was 5.78% out of the total number of births in the study period (2017–2021). The proportion of births with gestational diabetes mellitus was significantly increased in the COVID-19 period (2020–2021; 8.20%) compared with the preCOVID-19 period (2017, 2018, 2019; 4.06%). That could be explained by the anxiety during this period, reduced physical activities, or modified dietary habits (which can cause weight gain). The risk for gestational diabetes has been noted in women exposed to stressful conditions in the context of the COVID-19 pandemic, which can be the result of exposing pregnant women to many stressors that cause determined disturbances to their glucose metabolism. COVID-19 infection during pregnancy also increases the risk of maternal and fetal complications [25]. Two of the most important risk factors for gestational diabetes were analyzed in our study: BMI and age group. Obesity and maternal age are the two most important factors independently affecting the risk of GDM [10,11,26,27,28]. BMI is commonly used in risk-based screening for GDM. The prevalence of GDM increases with increasing BMI [29]. Those with a BMI between 30 and 40 have a 3.56 times higher risk, and those with a BMI over 40 have an 8.56 times higher risk of developing GDM [10,11,26,27]. Obesity is important risk factor for GDM [10,11,26,27]. In terms of obesity and diabetes mellitus, it has been shown that treatment with metformin in pregnancy can be useful in obese pregnant patients without diabetes because it significantly reduces weight gain and decreases the risk of preeclampsia and neonatal complications [30].

The proportion of mothers with GDM in our study was significantly increased among those with obesity (56.42%) compared to those without obesity (chi2 Fisher exact test, *p* < 0.001). The association between obesity and diabetes has also been shown to increase the risk of COVID-19 [31].

Advanced maternal age is an independent risk factor for GDM. Some studies [32,33] have suggested that GDM risk increases linearly with maternal age; another study showed that the incidence of GDM increased with age, peaked at 35–39 years, and then declined in women aged 40–50 years [8].

The proportion of mothers with GDM is significantly increased among mothers over the age of 30 [10,11,23]. In our study, the highest incidence of GDM was in the age group of 31–40 years (62.30%), while at >40 years, there were only 2 cases with GDM (1.64%). Mothers over the age of 30 represent a significant risk factor for the occurrence of GDM.

Fetal macrosomia is common in pregnancies with GDM; this is mainly due to the increased insulin resistance of the mother [1,34]. The extra glucose in the fetus is stored as body fat, causing macrosomia. The incidence of macrosomal fetuses in the literature is described as between 15% and 45% [1,34]. The prevalence of macrosomal fetuses in our group was 24.59%. The proportion of macrosomic newborns was significantly increased among mothers with GDM (25.41%) compared with mothers without GDM (3.85%) (chi2 Fisher exact test, *p* < 0.001).

An association between GDM among women and congenital malformations in their offspring has been suspected since the nineteenth century. According to various studies, the incidence of congenital malformations has not decreased over the past 25 years.

Diabetic embryopathy can affect any developing organ system, but cardiovascular and neural tube defects are among the most frequent anomalies. In our retrospective study, we found two cases of fetal malformations out of a total of 122 births of mothers with GDM, which represents 1.639%. Both fetal malformations were heart malformations. The incidence of fetal malformations among mothers with GDM is described to be around 6% [1,12,13,15].

There have been several studies on the neonatal outcomes associated with GDM and Apgar scores. A study of 94 patients with GDM showed lower 1 min and 5 min Apgar scores and increased incidences of perinatal morbidity of neonates compared to neonates of mothers without impaired glycemic control [35].

In the group of mothers with GDM (122 cases), we had 11 newborns with Apgar Score 6 (9.016%), 20 newborns with Apgar Score 7 (21.311%), and 85 newborns (69.872%) with Apgar Score 9.

There are no current absolute indications for elective C-sections in women with GDM; standard guidelines recommend C-sections when the estimated fetal weight is up to 4000 g [36,37]. It is important to identify which risk factors may lead to cesarean delivery in order to avoid possible complications for both mother and fetus. In our study, most pregnancies were delivered by C-section (86.89%). The indications for C-section were macrosomia (fetal weight >4000 g) and also fetal distress (reflected by Apgar Score).

### Strengths and Limitations of the Study

The study has some limitations: even though it covers a long period of time (5 years) and many births (17,230), we have a small number of patients diagnosed with gestational diabetes; additionally, the study includes patients from a single center. The costs related to performing the TTGO also limited us, and this was one of the reasons why we chose to do the test only in selected cases. That is why it might be interesting in the future to have a study that will take place in more centers with a larger number of pregnancies, where all pregnant women between 24 and 28 weeks will be tested. Another limitation of our study was that we did not test the glycosylated hemoglobin levels in our patients, only sporadically. The obstetric practice was limited by excessive rates of cesarean section.

The strength of this work is the focus on a pathology that can cause complications in pregnancies.

## 5. Conclusions

The study evidences the importance of the evaluation of all pregnant women with risk factors for developing GDM in order to establish an early diagnosis and treatment. Early detection of GDM and treatment at the appropriate time will reduce the social and financial burdens of managing the complications of diabetes. Maternal obesity and age more than 30 represent a significant risk of developing GDM. The prevalence of GDM during the COVID-19 period has increased, which can be explained by reduced physical activity, anxiety, or modified dietary habits (which can cause weight gain), even if the follow-up period was not impacted.

The higher prevalence of GDM during the COVID-19 pandemic can be the result of exposing pregnant women to many stressors that cause determined disturbances to their glucose metabolism. Metabolic complications specific to COVID-19 are not yet well characterized. However, additional research is needed to understand the relationship between physical activity changes, dietary patterns, and the increase in incidences of GDM during the COVID-19 period.

## Figures and Tables

**Figure 1 diagnostics-12-01241-f001:**
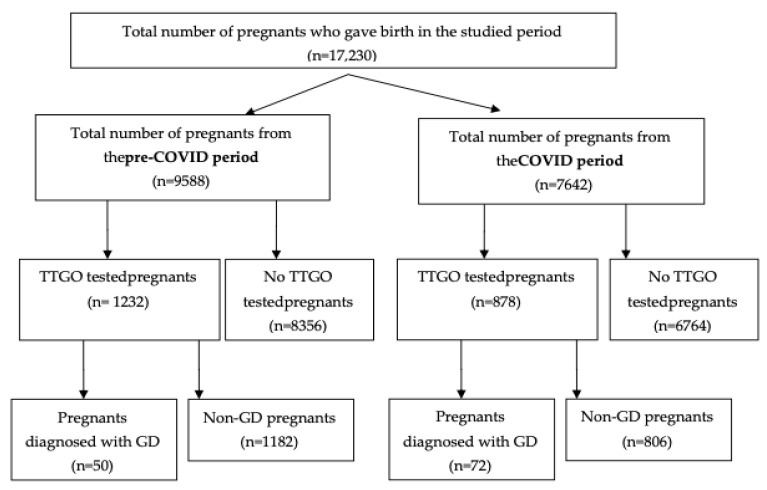
Flowchart describing patient selection; GDM: *gestational diabetes*
*mellitus*.

**Table 1 diagnostics-12-01241-t001:** The distribution per year of the prevalence of GDM births according to the total number of tested patients per year.

Total Births Per Year at Tested Patients		Total Births with GDM Per Year	% of GDM Births
2017	433	12	2.77
2018	447	13	2.91
2019	352	25	7.10
2020	371	29	7.82
2021	507	43	8.48

GDM—gestational diabetes mellitus.

**Table 2 diagnostics-12-01241-t002:** Maternal characteristics of the GDM group.

	Number	%
**Parity**		
Nulliparous	**65**	**53.27**
Pluripare	**39**	**31.96**
More than 2 births	**18**	**14.75**
**Age groups**		
≤20 years	0 (0.00)	
21–30 years	44	36.07
31–40 years	76	62.30
>40 years	2	1.64
**Obesity**		
Normoponderal	51	41.80
Grade I obesity	48	39.34
Grade II obesity	19	15.57
Morbid Obesity	4	3.29

Normoponderal (BMI = 18.50–24.99); Grade I obesity (BMI = 30–34.99); Grade II obesity (BMI = 35–39.99); morbid obesity (BMI > 40.00).

**Table 3 diagnostics-12-01241-t003:** Obstetric/neonatal outcomes of the GDM group.

	Number	%
**Type of birth**		
Natural Delivery	16	13.11
Cesarean Delivery	106	86.89
**Newborn weight at birth**		
<2000 g	3	2459
2000–2500 g	2	1.634
2500–3000 g	17	13.93
3000–3500 g	39	31.962
3500–4000 g	30	24.590
4000–5000 g	30	24.590
>5000 g	1	0.819
**Apgar Score (IA)**		
6	11	9.016
7	26	21.311
9	85	69.672
Malformed fetuses	2	1.639

Both the malformations mentioned above were heart malformations (VSDs). The median gestational age at birth was 36 weeks, and the median birth weight was 3200 g.

**Table 4 diagnostics-12-01241-t004:** The characteristics and results of the compared statistical tests.

*Variable*	*PreCOVID* *(n = 50)*	*COVID* *(n = 72)*	*p ^Test, sig^*
*HGB (g/dL)*	11.18 ± 1.48	10.92 ± 1.38	0.291 ^t, is^
*HCT (%)*	34.1 (7.44)	32.6 (6.7)	0.378 ^M-W, is^
*WBC (×10³/qL)*	12.1 (5.31)	11.5 (3.46)	0.306 ^M-W, is^
*Blood sugar level (mg/dL)*	103.0 (35.5)	101.5 (32.75)	0.613 ^M-W, is^
*Fetal weight (g)*	3400 (950)	3575 (912.5)	0.771 ^M-W, is^
*Apgar score*	9 (1)	9 (1)	0.213 ^M-W, is^
*HTAIS*	17 (34%)	19 (26.4%)	0.365 ^Chi2, is^
*Newborn pathology*	41 (82%)	50 (69.4%)	0.141 ^Chi2, is^

Legend: ^t^—unpaired *t*-test, ^M-W^—Mann-Whitney U-test, ^Chi2^—Chi2 test. ^is^—insignificant difference; HGB—hemoglobin; HCT—hematocrit; WBC—white blood cells; HTAIS—pregnancy-induced hypertension. *We obtained peripheral venous blood during fasting*.

## Data Availability

Not applicable.

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
