# Peer review of "Prevalence of Gestational Diabetes in preCOVID-19 and COVID-19 Years and Its Impact on Pregnancy: A 5-Year Retrospective Study"

_diagnostics, 2022, doi:10.3390/diagnostics12051241_

Round 1
Reviewer 1 Report
The article deals with the incidence and complications of gestational diabetes in one trietary centre in Europe. The authors show that behavioral changes induced by Covid-19 pandemia may have impact on higher incidence of GDM. The study is well planned and presented. I opt for publication without major corrections.
Author Response
Thanks for the review.
Reviewer 2 Report
Really interesting and well executed single centre study on the incidence of gestational diabetes in the Covid era and in the previous years.
I congratulate the authors for their effort. My suggestions and comments below:
- Discussion section: Speaking of obesity and diabetes mellitus it would be interesting to include a hint of the possible therapy in pregnancy (metformin). D'Ambrosio V et al. Metformin reduces maternal weight gain in obese pregnant women: A systematic review and meta‐analysis of two randomized controlled trials. Diabetes Metab Res Rev. 2019;35:e3164.
- It would be interesting (if you have the data) to also include how the incidence of gestational diabetes can be increased in mothers who contracted Covid in the 1-2 trimester of pregnancy)
Author Response
Thank you for your valuable comments.
- A paragraph about using Metformin as possible therapy in pregnancy was included;
- Unfortunately we have no sufficient data to debate this aspect but it is a good theme for a future research.
Reviewer 3 Report
The authors study the prevalence of gestational diabetes in a cohort of pregnant women presenting at a tertiary hospital in Romania.
One major concern is the low GDM rate. The authors obtain a GDM rate < 1% whereas other studies report a rate of 3-9%. The authors have provide additional clarifications such as what percentage of pregnant women are being tested for GDM in their region. Also, they should refer to similar studies conducted in their geographical region and compare their rate. The authors should discuss the potential bias introduced in their results by not screening the majority of their pregnant population.
Other comments:
- Re-organize Flow chart to show count of pregnancies before and after COVID pandemic, and then GDM pregnancies. This way Table 1, 3 and Figure 2 are not needed.
- Eliminate bullet points in lines 82-95
- Present Table 4 with the following columns (All pregnant women, All GDM, All pregnant women preCOVID, All GDM preCOVID, All pregnant women postCOVID, All GDM postCOVID) and study whether the population has changed pre versus post COVID.
- Figures 3, 4, 5 can be shown in Table 4 as explained above.
- The focus of the paper should not be on factors associated with GDM as this is a well discussed topic. The focus should be on the differences pre versus post COVID
- The authors can attempt to fit a model with outcome GDM and predictors (age+obesity) shown in Table 4, time linear (with a slope allowed to differ between pre and post COVID). This way they can answer whether the observed increase in GDM post COVID is still significant after allowing in change in factors and the observed increase in GDM over years.
- If parity or singleton deliveries are available please show in Table 4. You could show in Table 4 all 3 values of OGTT (fasting, 1h, 2h) and count of individual mothers. I assume if a mother had more than one delivery she was included twice in the analysis.
- Table 4 could be split into two tables: maternal characteristics and obstetric/neonatal outcomes.
- Each obstetric/neonatal outcomes can be modeled using predictors GDM, age, obesity, time as explained before.
Author Response
Thank you for your valuable comments .Unfortunately we have no centralised database in the region. There were no similar studies in the region as we know. For a future research is a good idea,
- Points 1 and 2 were modified after the recommendations.
- The aim of the study was to determine the incidence of GDM, to lay out the risk factors associated with GDM and also observe the evolution of pregnancy among patients with this pathology by emphasizing the state of birth of the fetus, the birth weight and the way of birth. Therefore we didn’t record the data from all the patients whom delivered in the studied period. It is a good theme for a future research.
- The figures 3 ,4, 5 were deleted.
- We complete Table 4 with information related to parity and singleton deliveries.
In the GDM group no patient deliver more than once.Table 4 was split in two tables.
Reviewer 4 Report
Reviewer
This paper aimed to determine the incidence of GDM and to lay out the risk factors associated with GDM in preCOVID-19 and COVID-19 yeas and also observe the evolution of pregnancy among patients with this pathology by emphasizing the state of birth of the fetus, the birth weight and the way of birth
INTRODUCTION
- Are there any reports that compared the difference of anxious, physical activities and modified dietary habits of individuals in preCOVID19 and COVID19 period to support the rational of this study?
- Are the medical behavior changes in individuals (to see doctor in clinic vs. medical centers) in preCOVID19 and COVID19 period?
METHODS and results
- Multiple logistic regression method for preCOVID19 and COVID19 to predict GDM should be used?
- The results of Table 3 and Fiugre2, Table 4 and figure 3,4 seem the same and duplicated.
- Table 10? (Maybe Table5).
DISCUSSION
- The results of this study to make conclusions that Covid years is a risk factor in developing gestational diabetes are not sufficient.
- There is selection bias for GDM before and after covid19 years because many clinics may decrease the patients in Covid19 years to prevent Covid19 infection. Therefore, more pregnant women including complicated women can only visit in hospital. The confounding factors of changes in medial behavior should consider in this study.
Author Response
Thanks for the review
- Unfortunately, we do not have any reports regarding the anxious physical activities and the modified eating habits during the preCovid and Covid period.
- The statement was based on anamnestic data that revealed that pregnancy during the covid period was a matter of concern for the mother due to the exact ignorance of the risk that Covid 19 infection may have on the evolution of the pregnancy and the fetus.
- Figures 3 and 4 have been deleted from the text.
Round 2
Reviewer 3 Report
GDM prevalence of 0.7% is strikingly low as compared to other studies. This raises questions about the quality of the data. It is not clear whether the entire study cohort (n=17230) was screened for gestational diabetes. The quality of the data and the potential sources of bias has to be understood before making conclusions.
Author Response
Dear Reviewer,
Thanks for the feedback. We identified the fact that the colleague who dealt with the statistical part made a wrong reference to the total number of patients who gave birth during the study period. But only some of them were tested for GDM. Therefore, the statistical part that made the comparison with all the patients who gave birth was redone. We made the corresponding changes in all sections of the paper, reporting the data only to the patients who performed TTGO in the current pregnancy. The screening methodology and the gestational age range at which TTGO was applied were completed.
We made some changes to the discussions including some limitations of the current study.
Reviewer 4 Report
The revised version has corrected according to the previous suggestion. No more comments are required.
Author Response
Dear reviewer,
Thank you for your support.
Sincerely,
Mrs. Dana Chiriac
